# Diagnosis and Mitigation of Electromagnetic Interference Generated by a Brushless DC Motor Drive of an Electric Torque Tool

Jerzy Baranowski *, Tomasz Drabek, Paweł Piątek and Andrzej Tutaj

Faculty of Electrical Engineering, Automatics, Computer Science and Biomedical Engineering, AGH University of Science and Technology, Al. A. Mickiewicza 30, 30-059 Krakow, Poland; drabek@agh.edu.pl (T.D.); ppi@agh.edu.pl (P.P.); tutaj@agh.edu.pl (A.T.)
* Correspondence: jb@agh.edu.pl

**Abstract:** Electrical devices in the consumer markets need to comply with stringent standards for electromagnetic interference (EMI) distortion and electromagnetic compatibility (EMC). This paper presents the results of measurements of electromagnetic interference generated by an electrical drive of an electric torque tool with a brushless DC motor. The measurements were made in accordance with the PN-EN 55014-1:2017-06E standard, in the frequency band of 148 kHz–30 MHz. The results confirmed that the tested drive can meet the requirements defined in this document. Another document, the PN-EN IEC 61000-3-2:2019-04E standard, provides limits for the harmonic content in the current drawn by electrical devices from a single-phase AC line. This paper also presents the results of measurements related to this standard. Harmonics, up to and including the 40th one, were determined and compared with the limits given in the standard for class B devices. The measurement results indicate a need to use an active power factor corrector (PFC) filter. Such a system has been tested by computer simulations. The results confirmed its ability to meet the requirements of relevant standards.

**Keywords:** brushless DC motor; electronic commutation; EMI; EMC; disturbance diagnosis; higher harmonics mitigation; disturbance mitigation

## 1. Introduction

The basic legal acts enabling the proper functioning of European Union markets are Directives. One of them is Directive 2014/30/EU [1] (called the Electromagnetic Compatibility (EMC) Directive), related to the electromagnetic compatibility of devices. The technical side of this document contains references to harmonized standards in the field of EMC. In accordance with the recommendations contained in Annex I to the Directive [1], devices should be designed and manufactured in such a way that the electromagnetic interferences generated by them do not exceed the level above, in which they could prevent nearby radio and telecommunications devices or other devices from operating as intended. The Directive [1] divides electromagnetic environments into two basic classes for which different limits of emission have been established:

- residential, commercial, and slightly industrialized and
- industrial.

Since the 20th of April 2016, when the latest version of the EMC Directive was enforced, a new document confirming compliance has been used. The EC Declaration of Conformity was replaced by the EU Declaration of Conformity. Directive 2014/30/EU allows the self-assessment of products, i.e., without the participation of a notified body.

Consumer electrical appliances (household appliances) and power tools are directly addressed by the PN-EN 55014-1:2017-06E standard [2], in which the permissible levels of electromagnetic disturbances are provided, and the methods of disturbance measurements

are described. In accordance with the EU requirements, every device offered on EU markets should undergo a conformity assessment procedure and have an electromagnetic compliance declaration. In the case of devices subject to the EMC Directive, one of the recommended methods is to perform device tests confirming compliance with harmonized EMC standards. For consumer electrical devices and power tools, the standard [2] is the basic document.

This paper presents the results of measurements of electromagnetic conducting disturbances generated by an electrical drive designed for an electric torque tool with a brushless DC motor. It also discusses the mitigation methods employed by the authors. The embedded drive controller implements a complex master algorithm responsible for motion trajectory generation, tightening, torque monitoring, and limitation to a preset value, process stage, and operation mode determination and switching, as well as diagnostic procedures and safety functions. The algorithm is implemented as a bare metal C language application running on a modern STM32 family microcontroller. The microcontroller also implements the low-level control of a permanent magnet brushless DC motor, drives a three-phase bridge of an intelligent power module responsible for commutation and modulation, and controls the current drawn by the motor. The drive with its controller is a source of electromagnetic interference (EMI), which, according to EU regulations, must not exceed allowable limits if the product is to be put into production and brought to the market [3,4]. This is a serious technical problem, often underestimated and neglected in scientific activities.

The standard PN-EN IEC 61000–3-2:2019-04E [5], not mentioned earlier, defines limits for the harmonics content in the current consumed by an electrical device from a single-phase AC line. Therefore, the paper also presents the results of measurements related to this standard. Harmonics, up to and including the 40th one, were determined and compared with the limits given in reference [5] and applicable to the devices of class B defined in it. The problem of generating disturbances by working brushless DC (BLDC) motors and the methods of designing inverters to reduce disturbances were described in the works of references [6,7]. Interesting methods of reducing the generated disturbances were also proposed in the works [8–10].

Initially, the research concerned an electric drive without any EMC filter or a power factor corrector (PFC), due to the limited space in the power tool housing. The aim of the tests was to check whether, without these systems, the drive meets the standard requirements regarding the generated conducted electrical interferences. Since the limits defined by both standards have been violated, some supplementary circuits have been proposed to mitigate the problem. The effectiveness of the adopted measures was partially verified by experiments and partially by computer simulations.

The rest of the paper is organized as follows. First, we describe the device under test, which is a BLDC motor used for an electric torque tool, along with testing procedure. Then, we provide the test results and extensive analysis of their consequences, and we propose a remedial for situation supplemented with a simulation analysis. The paper ends with the conclusions.

## 2. Materials and Methods

### 2.1. Device under Testing

The device under testing was an electric drive system with a brushless DC motor (BLDC), developed using an electric torque tool with a rated torque of 1000 N·m (Figure 1) [11,12]. It is intended as a replacement of a drive with a commutator AC motor used in the previous tool version. The rated data of the motor are as follows: the output power is 1.45 kW, the rotational speed equals 10,000 rpm, the torque is 1.38 N·m, the frequency of back emf voltage is equal to 666.6 Hz, and the RMS (root mean squared) value of the phase current is 6.2 A. The motor has 8 poles (4 pole pairs), and the drive is supplied with the RMS voltage of 230 V AC. The input rectifier bridge is a 4-diode Graetz bridge. Capacitive filter and energy storage are a battery of electrolytic capacitors with a total

capacity of 1100 µF together with a 1-µF MKP (Metallisierter Kunststoff Polypropylen) capacitor. The inverter bridge is a six-transistor three-phase intelligent power module (IPM) STGIF10CH60TS with IGBT (insulated-gate bipolar transistor) transistors. The control system based on the STM32F407VGT6 microcontroller performs six-step commutation and pulse width modulation (PWM) [13]. An EMC input filter 10EMC3 F8148 1639 R 2 from CORCOM, with the internal structure shown in Figure 2, was used as an optional disturbance mitigation device. The filter parameters, given on its rating plate, are as follows: 10 A, 120/250 V, 50–60 Hz, −10–+40 °C, L-2 × (0.74 + 0.74) mH, C-2 × 1 µF (X2) SH, 2 × 2.8 nF (Y), R-2 × 470 kΩ.

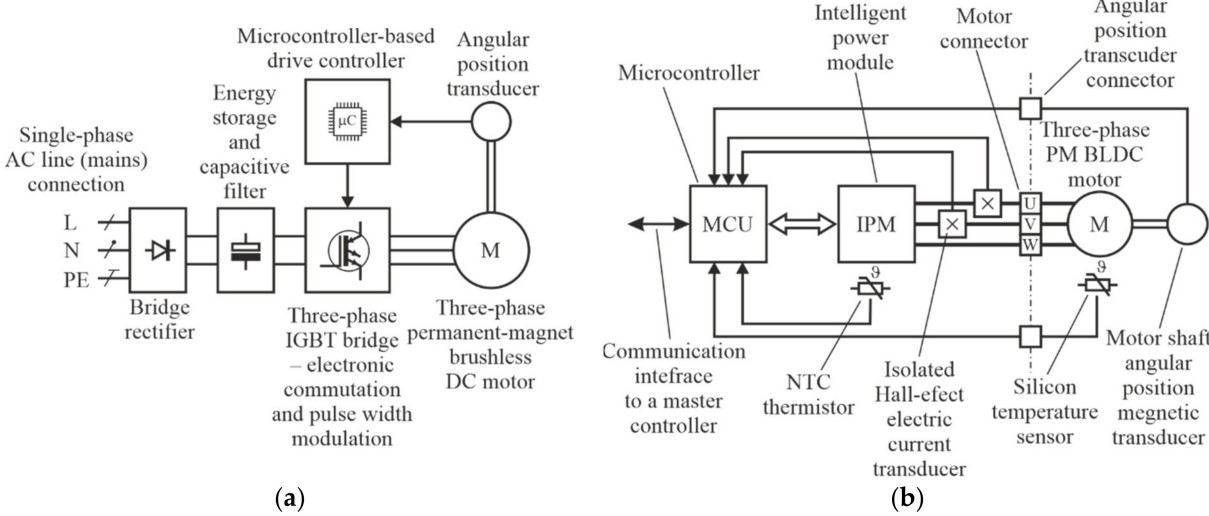

(**a**)  (**b**)

**Figure 1.** Key components of the permanent magnet brushless DC motor drive system. (**a**) High-current circuit structure. Power stage acts as a commutator and a pulse width modulation (PWM). The PWM can be a source of high-frequency disturbances. This structure lacks a power factor corrector (PFC) filter that would mitigate higher harmonics. (**b**) Control system structure. It contains monitoring sensors for overheating and overcurrent detection. Each of the transistors in the intelligent power module (IPM) is independently controlled by the microcontroller unit (MCU). The IPM takes care of some safety constraints.

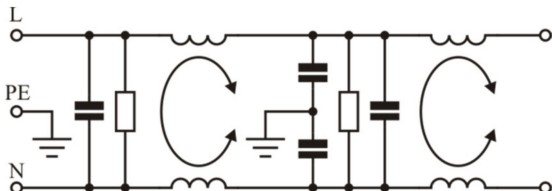

**Figure 2.** An optional input electromagnetic compatibility (EMC) filter can be connected at the output of the drive. We used the EMC input filter 10EMC3 F8148 1639 R 2 from CORCOM. (L-line, PE-protective earth, N-neutral)

To regulate the motor speed, a classical control system was used, consisting of a master PI (proportional-integral) speed controller and a slave PI controller for the motor current. The motor is powered by a 6-transistor IGBT bridge, which implements the pulse width modulation (PWM) of the motor phase voltage. In such a system, the PI current controller adjusts the PWM duty factor based on the measurement of the average direct current (input current) of the IGBT bridge. The dynamics of the applied speed and torque control system meet the utility requirements of the power tool. Its computational implementation does not overload the STM32 microcontroller, whose computing power is also needed to implement the master torque control of the power tool during the entire screwing and tightening process and for other purposes. For these reasons, advanced motor speed and current control algorithms, such as those proposed in [14–17], were not used.

## 2.2. Test Scenarios and Test Bed

Tests conducted according to the standard [2] encompassed measurements of the continuous network disturbance voltage in the 148-kHz–30-MHz frequency band, using a complete drive system consisting of a rectifier, six transistor commutation and modulation bridges, a microprocessor-based drive controller, and a PM brushless DC motor for the following conditions:

- at the idle run condition,
- at the rated motor load torque, with and without PWM motor phase voltage modulation,
- for different engine speeds corresponding to different voltages supplying the entire drive (commutation only, no PWM modulation), and
- in all cases mentioned above—for both possible directions of engine rotation.

All tests were carried out using the measuring system shown in Figure 3. A HM6050 2 type V line impedance stabilization network (LISN) (50 Ω / 50 µH + 5 Ω) manufactured by HAMEG, compliant with CISPR 16, was used in the circuit. The LISN was supplied from a 230-V/230-V isolation transformer with a split secondary winding (2 × 115 V) and was connected to a HMS X Rohde & Schwarz spectrum analyzer. The isolation transformer was powered from a sinusoidal alternating voltage source separated from the power grid. A three-phase synchronous generator with a rated power of 27 kVA was employed, and its phase-to-phase voltage was used to power the transformer. The generator was driven by a separately excited shunt DC brushed motor fed by an AC/DC converter. The generated voltage met the requirements of the standard [5] in terms of the shape and stability of the RMS value and frequency of the voltage supplying the measuring system.

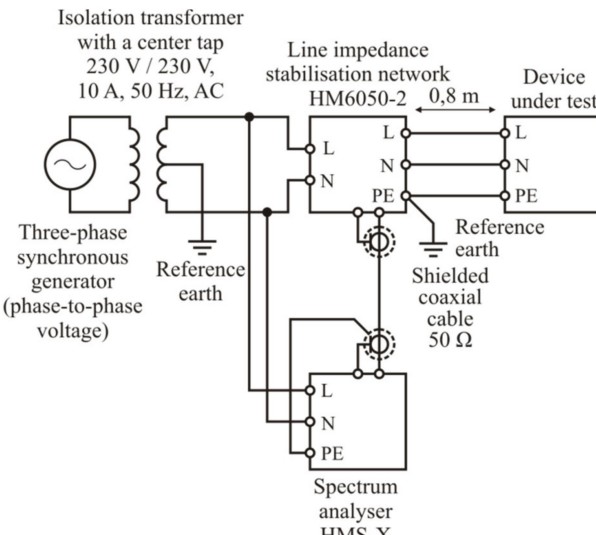

**Figure 3.** Simplified schematic diagram of a test circuit used for measuring the conducted interference in the 148-kHz–30-MHz frequency band. Our system includes a three-phase generator, in which we used phase-to-phase voltage. It is connected to an isolation transformer with a grounded central tap. A spectrum analyzer is powered from the transformer and process measurements from a line impedance stabilization network (LISN) network loaded with our device under testing.

1. The tested BLDC motor was mechanically loaded by another BLDC machine acting as a generator. In turn, the generator was electrically loaded with high-power resistors via a three-phase diode rectifier and an electronic chopper. The chopper controlled with a PWM signal was used to vary the load torque and power. The torque was measured with a rotary torque transducer. The rotational speed was inferred from the frequency of the generator voltage. The mechanical power was computed based on the torque and speed measurements.

2.  The following requirements defined by the standards and regulations were observed in the construction of the test bed:
3.  Required minimum distance between the elements of the measuring system and its surroundings:

    (a)  A distance of 0.8 m between the artificial network (LISN) and the test object.
    (b)  A distance of 0.8 m between the transformer supplying the entire system and each of its elements, due to the magnetic leakage flux of the transformer windings.
    (c)  A distance of 0.8 m between any element of the measuring system (including the device under test) and any grounded surface, including the walls of the room.

4.  The drive protective conductor, which is a physically an additional conductor, is routed along the power cables at a distance not exceeding 0.1 m.
5.  The signal connection between the spectrum analyzer and the artificial network was made according to Figure 1, with a 1.5-m-long coaxial cable of 50-Ω wave impedance.
6.  The measuring table was placed on a so-called reference earth, i.e., a 2-m by 2-m rectangular, grounded, conductive metal plate.
7.  Connections between the reference ground point, the isolation transformer, secondary winding center tap, and the PE (protective earth) terminal of the artificial network were made in a way that allows the electric currents of the tested frequencies to flow into the reference ground. For this purpose, the artificial network was placed directly on the reference earth plate so that the metal housing of the artificial network system (conductive and connected to the PE terminal of the device) was in contact with the reference ground on the largest possible surface. The connection of the transformer winding center point with the reference ground was made with a wide, multi-stranded braided copper tape.

An isolation transformer, not formally required, was used to reduce the radio and electrical interferences penetrating from the synchronous generator into the measuring system. To check the correctness of the measuring station operation, preliminary measurements were carried out in a complete measurement system but without the test object (i.e., with an open LISN load circuit)—see Figure 4. In all the performed tests, the quasi-peak values of the disturbances were measured.

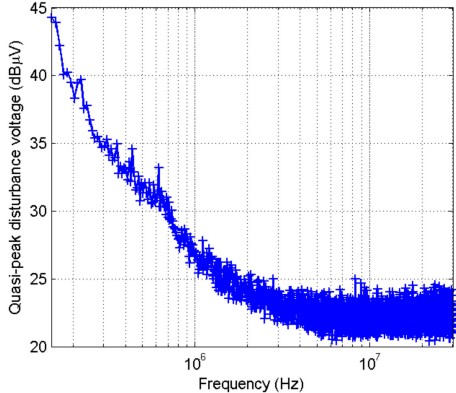

**Figure 4.** Testing for the conducted EMI. Reference measurement results for a test circuit with no device under the test connected (LISN network unloaded, network output open). All noises are well below the 45-dBμV level, which we consider as the background noise floor.

### 3. Results

The results of measurements of continuous conducted interference in the 148-kHz–30-MHz band are presented in Figures 5–8. According to the standard [2], the allowable level of the quasi-peak interference voltage is from 76 to 69 dBμV in the frequency range

148 kHz–350 kHz, decreasing linearly with the logarithm of the frequency 69 dBμV in the range 350 kHz–5 MHz and 74 dBμV in the range 5–30 MHz. In Figures 5 and 6, these limits are marked with a solid green line. The characteristics in Figures 7 and 8 are entirely in the allowed region, so no limit lines were needed. The results of the measurements of the harmonics of the current drawn by the drive system for frequencies up to 2000 Hz (up to the 40th harmonic component, according to reference [5]), are presented in Figures 9–12. The logging of the drive current waveforms was conducted in the system shown in Figure 3, using an additional Hall effect clamp current probe with a declared frequency range up to 200 kHz (−1 dB) and a Rhode & Schwarz digital storage oscilloscope (DSO). The EMC filter tested earlier was used in all tests. The conditions of individual tests are given in Table 1. The admissible RMS values of individual harmonics of the drive mains current for a class B device, according to the standard [5], are marked in Figures 9–12 with a green solid line.

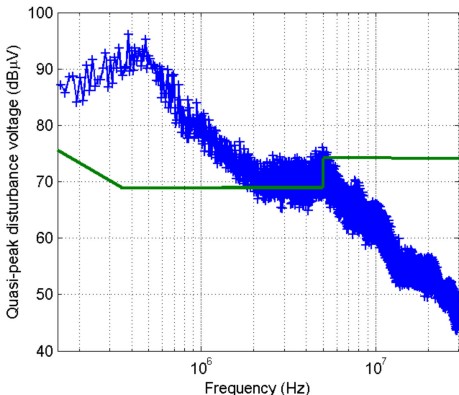

**Figure 5.** Testing for the conducted EMI. Measurement results for the no-load motor condition, no PWM, at the rotational speed of $n$ = 5000 rpm. Speed is controlled by adjustment of the RMS value of the AC voltage feeding the drive. No input EMC filter. Green line denotes the standard imposed levels. The considered system exceeds them significantly up to 1 MHz.

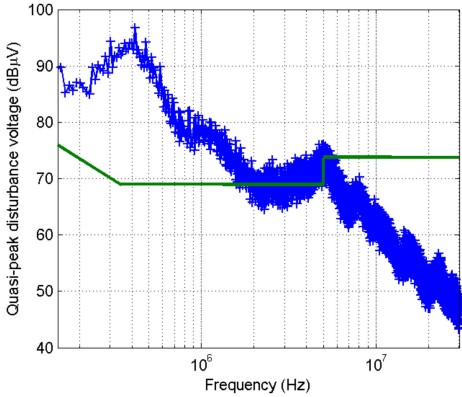

**Figure 6.** Testing for the conducted EMI. Measurement results for the motor loaded with the rated torque, no PWM modulation, at a speed of $n$ = 10,000 rpm. Speed controlled by adjustment of the RMS value of the AC voltage feeding the drive. No input EMC filter. Green line denotes the standard imposed levels. Additional load torque and increasing rotational speed does not change the high frequency behavior.

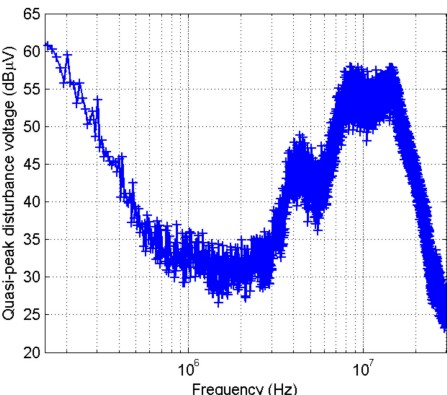

**Figure 7.** Testing for the conducted EMI. Measurement results for the motor loaded with the rated torque, with no PWM modulation, at the speed of *n* = 5000 rpm, with the input EMC filter. Speed controlled by adjustment of the RMS value of the AC voltage feeding drive. Adding an EMC filter reduces the noise to the standard accepted levels.

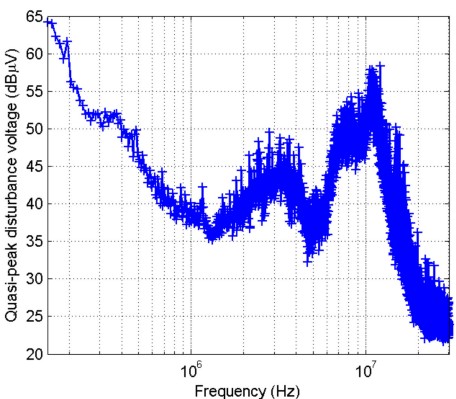

**Figure 8.** Testing for the conducted EMI. Measurement results for the motor loaded with the full rated torque, with PWM modulation, at the speed of *n* = 5000 rpm, with an input EMC filter. The drive was fed with the full AC line voltage of 230 $V_{RMS}$, 50 Hz. The introduction of a PWM control increases the noise levels to around a 100-kHz frequency, but the rest is comparable to the nonmodulated control case.

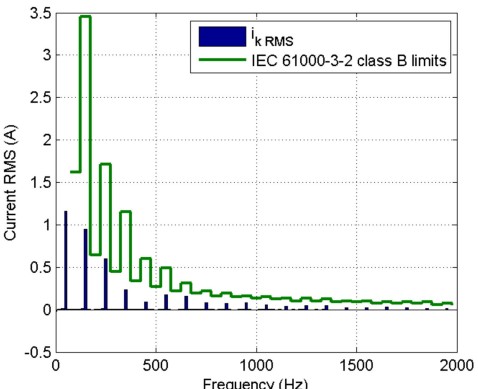

**Figure 9.** Testing for harmonic current emissions. RMS values of the harmonic components of the current drawn by the tested drive for idle motor conditions at *n* = 10,000 rpm and for the PWM duty cycle of 67.7%. The duty cycle was chosen for the rated speed. All higher harmonics are well below standard defined limits. This was to be expected, as the current is drawn from the mains while the tool is idling is relatively low.

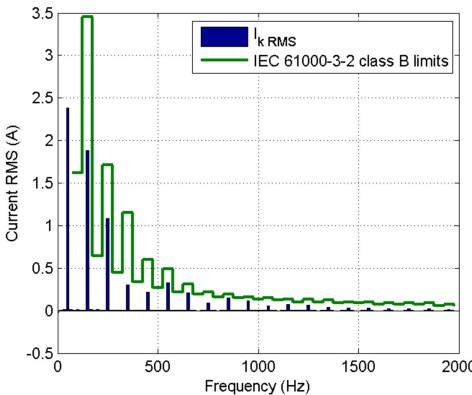

**Figure 10.** Testing for harmonic current emissions. RMS values of the harmonic components of the current drawn by the tested drive for the motor loaded with half the rated torque at *n* = 5000 rpm and for the PWM duty cycle of 40%. A moderated load and speed operation does not cause the current harmonics to exceed the standard limits.

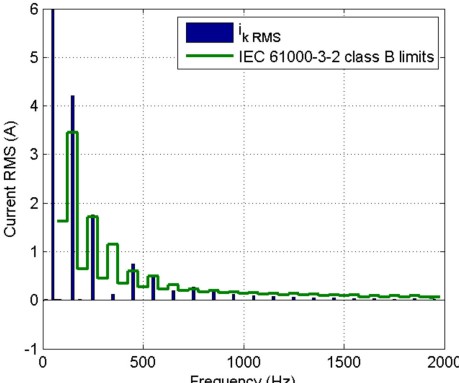

**Figure 11.** Testing for harmonic current emissions. RMS values of the harmonic components of the current drawn by the tested drive for the motor loaded with the full rated torque at *n* = 10,000 rpm and for the PWM duty cycle of 81%. In full-rated operations, the harmonics exceed the limits.

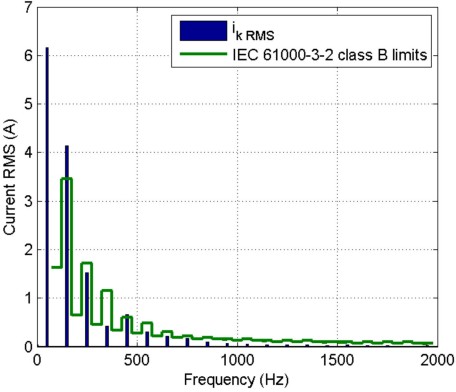

**Figure 12.** Testing for harmonic current emissions. RMS values of harmonic components of the current drawn by the tested drive for the motor loaded with the full rated torque at *n* = 10,000 rpm and for the PWM duty cycle of 81%. Two chokes of 0.68-mH inductance are placed on the L and N lines, respectively, between the input EMC filter and the Graetz bridge rectifier. This modification did not improve the standard compliance.

**Table 1.** Conditions of the tests conducted to measure the harmonic components of the current drawn by the drive system under testing. PWM: pulse width modulation.

| Test Run | Motor Load Torque | Motor Rotational Speed | PWM Modulation Duty Cycle | Additional Chokes |
|---|---|---|---|---|
| Idle (Figure 9) | 0 Nm | 10,000 rpm | 67.7% | no |
| Second (Figure 10) | 0.69 Nm | 5000 rpm | 40% | no |
| Third (Figure 11) | 1.38 Nm | 10,000 rpm | 81% | no |
| Fourth (Figure 12) | 1.38 Nm | 10,000 rpm | 81% | 2 × 0.68 mH |

## 4. Discussion

The results of the harmonic analysis of the drive supply current indicate that even the use of additional DEHF 42/0.68/13 HD22 FERYSTER (2 × 0.68 mH) chokes in the power lines does not allow to meet the requirements of the standard [5] in all drive operating conditions. Therefore, in the production version of the drive, it will be necessary to use an active power factor corrector (PFC) system with a dedicated controller [18,19].

A model of such a corrector, intended for the tested drive, built with the MATLAB/Simulink/Simscape/Simscape Electrical/Specialized Power Systems toolbox, is shown in Figure 13. The corrector should be incorporated into the drive system presented in Figure 1 by placing it between the bridge rectifier and the energy storing capacitor. The heart of the corrector is a step-up (boost) voltage converter, consisting of a choke (L = 1.36 mH), a MOSFET power transistor acting as a key, a fast rectifier diode, and a control system [20]. It is powered by the mains voltage rectified by a Graetz bridge and supplies energy to the load (here, represented by a resistor, R = 80 Ω) and its bypass capacitor (C = 1100 µF).

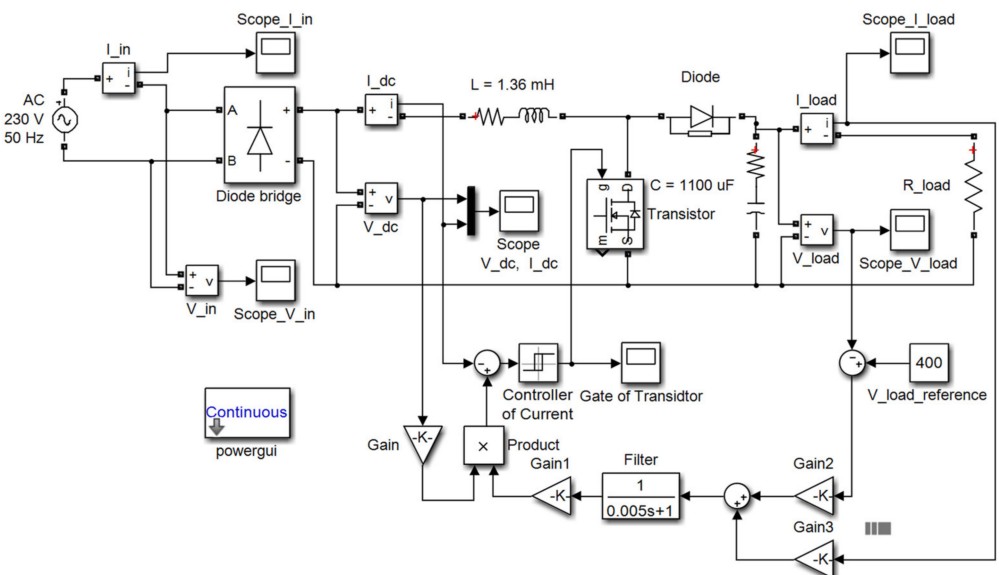

**Figure 13.** Simplified simulation model of the active power factor corrector filter built in the MAT-LAB/Simulink/Simscape/Simscape Electrical/Specialized Power Systems environment. The filter employs a boost converter architecture with a single choke, a MOSFET power transistor, and a rectifier diode. The transistor is keyed by a controller for the current drawn from the AC line to follow the sinusoidal line voltage profile. The setpoint profile is scaled by the output signal of a feedback feedforward controller. The controller keeps the storage capacitor DC voltage at 400 V (above the AC line peak voltage) and adjusts the control signal based on the load current.

The inverter key is controlled by a two-level hysteresis controller of the current drawn from the bridge. Symmetric on- and off-switching thresholds for the current error are equal to ±0.5 A, respectively. The current setpoint waveform is calculated as a product of the normalized rectified AC mains voltage and a low-pass-filtered output signal of a load voltage regulator. It is a proportional feedback controller, and its 400 V setpoint was chosen to be higher than the AC line peak voltage (325 V). Since the controller gain is 0.246, a

10-V error in the bypass capacitor voltage results in a 1.739-A RMS current drawn from the mains. The voltage regulator, working in a closed feedback loop, was supplemented with a proportional feedforward control element, whose output signal depends on the value of the current consumed by the load. The equivalent gain in the feedforward path was set to 2.5825 to make the average value of the power drawn from the AC line slightly greater than the power consumed by the load of the PFC corrector. The power surplus is expected to cover the power losses in the choke and semiconductor components. The sum of the feedback and feedforward control signals is filtered with a first-order lowpass filter with a time constant T = 5 ms. Its purpose is to attenuate the control signal fluctuations resulting from ripples of the bypass capacitor voltage and the load current and to prevent them from affecting the AC line current waveform.

In the simulation model, the electrical drive with a brushless DC motor was replaced by a resistive load, because in the steady-state conditions, the load characteristic hardly affects the waveform of the current drawn from the AC network. For a load of R = 80 Ω, the average load current is about 5 A, the average load voltage is around 400 V, and the load power equals approximately 2 kW. The corresponding current drawn from the network (I_in) is shown in Figure 14, together with the network voltage (V_in). The results of the harmonic analysis of the AC line current, presented in Figure 15, show that the drive with the PFC system may be able to meet the requirements of the standard [5]. The total harmonic distortion (THD) factor of the current equals 3.81%, and the largest (third) higher harmonic does not exceed 2.5% of the fundamental harmonic current.

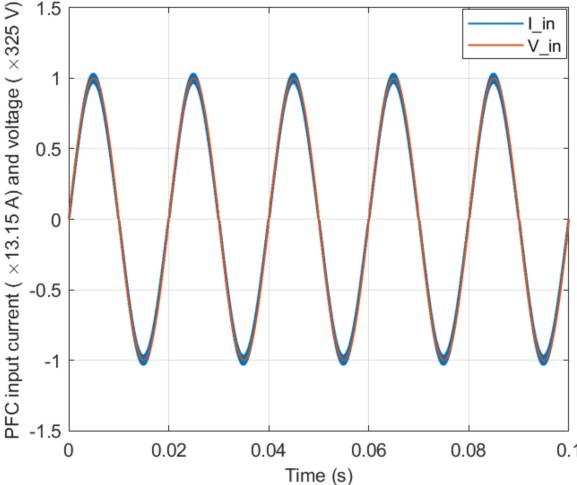

**Figure 14.** Time series of the voltage (red) and current (blue) drawn from the AC line at the steady-state conditions for the load power of *p* = 2 kW. Analysis of the PFC controller behavior in the time domain shows that a sinusoidal current setpoint profile is followed with a phase shift less than 2 degrees and limited current ripples.

The switching frequency of the power electronic key driven by the hysteresis current controller in the considered PFC corrector fluctuates between 3.5 kHz and 73 kHz. One can reduce the frequency by choosing a higher inductance choke. However, that would considerably increase the weight of the system. In the case of a handheld tool, this may be a prohibitive factor. Hence, it is recommended to keep the switching frequency unaltered and to employ a modern silicon carbide (SiC) active power electronic device that is able to handle switching the frequencies of several tens of kHz.

A harmonic analysis was performed for several different load resistances covering the range from 50 Ω to 200 Ω. Selected results are provided in Table 2. A relatively low content of higher current harmonics confirms that the PFC corrector is able to effectively mitigate the current distortion problem while maintaining a constant voltage across the energy storage capacitor.

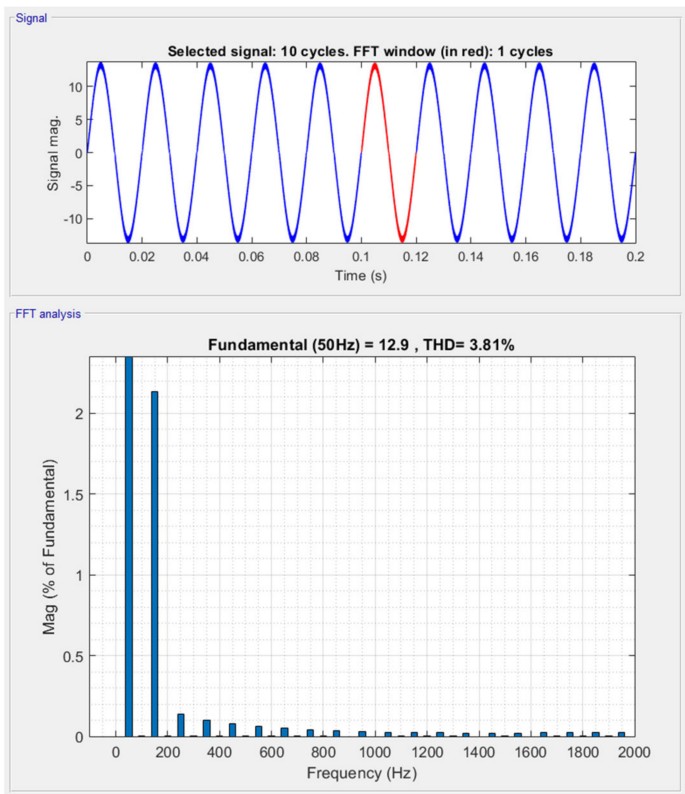

**Figure 15.** Results of the harmonic analysis for a single period (marked red) of the current drawn from the AC line by the simulated circuit at *p* = 2 kW. Total harmonic distortion was reduced to 3.81%, and the results are promising for ensuring a standard compliance.

**Table 2.** Performance of the power factor corrector (PFC) for various load resistances. The following quantities are provided: the load power, the average load current, the average load voltage, the total harmonic distortion (THD) factor for the AC line current, and the ratio of AC line current harmonics—the third to the first. All simulation results correspond to the steady-state conditions.

| Load Resistance | Average Load Current | Average Load Voltage | Average Load Power | THD Factor | Ratio of Third and First Harmonics |
|---|---|---|---|---|---|
| 200 Ω | 1.998 A | 399.5 V | 798.2 W | 7.99% | 2.50% |
| 170 Ω | 2.351 A | 399.7 V | 939.8 W | 7.00% | 2.36% |
| 140 Ω | 2.857 A | 399.9 V | 1142 W | 5.91% | 2.32% |
| 120 Ω | 3.334 A | 400.1 V | 1334 W | 5.20% | 2.27% |
| 100 Ω | 4.004 A | 400.4 V | 1603 W | 4.49% | 2.20% |
| 90 Ω | 4.450 A | 400.5 V | 1783 W | 4.16% | 2.20% |
| 80 Ω | 5.009 A | 400.7 V | 2008 W | 3.81% | 2.14% |
| 70 Ω | 5.729 A | 400.9 V | 2297 W | 3.48% | 2.10% |
| 60 Ω | 6.687 A | 401.2 V | 2684 W | 3.14% | 2.05% |
| 50 Ω | 8.031 A | 401.6 V | 3226 W | 2.78% | 1.96% |

## 5. Conclusions

The paper addressed the problem of current harmonic distortion and electromagnetic interference generation by an electronic drive system with an electric torque tool. The performed research resulted in the following observations:

An increase in the load experienced by the complete converter system results in a relatively low increase in the noise level in the 148-kHz–30-MHz band, as does a change in motor speed.

The introduction of PWM in a drive power stage causes a considerable increase in the level of interference in the 148-kHz to 30-MHz band but only for certain specific frequencies associated with the modulation frequency.

The application of an EMC filter on the 230-V AC power supply side brings a significant reduction in the noise level in the 148-kHz–30-MHz band. With the EMC filter, the drive meets the requirements of the standard [1] for this frequency band. Even when the drive is loaded with the full rated torque, the measured voltage levels are lower than the permissible values for all frequencies.

The research has shown that the amplitude of the motor phase current fluctuations has an impact on the magnitude of the conducted disturbances in the 148-kHz–30-MHz band. The measured disturbances also show the effects of motor phase commutation. Therefore, it is important to reduce the amplitude of the current fluctuations and to properly conduct the commutation of the phases. The phase current control algorithm affects the conducted disturbances only if it allows large phase current fluctuations. The main cause of fluctuations in the instantaneous values of the phase currents and the motor torque is the limited switching frequency of the IGBT transistors (20 kHz). Hence, the conducted disturbances in the band of 148 kHz–30 MHz can be mitigated by increasing the PWM frequency to 40–60 kHz. This requires replacing the IGBT transistors with SiC devices.

The effective values of the harmonics of the current drawn by the drive do not meet the requirements of the standard [5]. The insertion of two additional 0.68-mH chokes into the L and N power lines did not bring sufficient improvement. Limitations on the size and weight of the electric torque tool make it impossible to use chokes with higher inductance. An alternative solution to this problem can only be to employ an active PFC system, placed after the rectifier bridge (in a DC circuit). The simulation tests of the PFC system showed its full functional usefulness, both in reducing the content of higher harmonics in the AC line current and in providing the power factor value (cos φ) close to one for the complete drive. As part of a further work, it is planned to implement an active PFC filter and test the complete drive system under nominal operating conditions.

The PWM modulation frequency of 20 kHz is relatively high compared to the frequencies of the first 40 harmonics of the drive input current. Hence, even in the case of a large amplitude of the motor phase current fluctuations (at $n > 3000$ rpm), the PWM modulation does not affect the amplitude of the AC line current harmonics. Their values result from the operation of the converter input system, i.e., a four-diode rectifier with a large electrolytic capacitor behind it. Neither increasing the PWM frequency nor changing the motor phase current control method will cause the harmonic amplitudes of the input current to drop to the required values.

Research has shown that the use of an EMC filter and a PFC system is necessary. From an application point of view, the main limitation for the PFC corrector design is imposed by the highly restricted space in the housing of the power tool. This especially affects large-size electrolytic capacitors, e.g., C = 1100 μF and chokes. It can be difficult to cool the PFC power transistor for the same reason. Apart from these constraints, there seems to be no other limitations to the application of the PFC in the considered power tool.

**Author Contributions:** Conceptualization, J.B., T.D., P.P. and A.T.; methodology, T.D.; software, A.T.; validation, J.B.; investigation, P.P. and A.T; writing—original draft preparation, T.D.; and writing—review and editing, J.B., P.P. and A.T. All authors have read and agreed to the published version of the manuscript.

**Funding:** This work was partially funded from the project titled "Brushless Electric Torque Tool", which was financed by the National Centre for Science and Development under contract no. PBS3/B4/13/2015 and partially from AGH University of Science & Technology subvention for scientific activity.

**Institutional Review Board Statement:** Not applicable.

**Informed Consent Statement:** Not applicable.

**Data Availability Statement:** The data presented in this study are available on request from the corresponding author.

**Acknowledgments:** We would like to thank Tomasz Dziwiński for his help, and ZBM OSSA company for providing us with the motor.

**Conflicts of Interest:** The authors declare no conflict of interest.

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
