# Peer review of "Diagnosis and Mitigation of Electromagnetic Interference Generated by a Brushless DC Motor Drive of an Electric Torque Tool"

_energies, doi:10.3390/en14082149_

Round 1

Reviewer 1 Report

Please see the attaeched file.

Reviewer 2 Report

The submission is more of a test report than a scientific article paper. There is little theoretical innovation. Some other concerns include: Insufficient literature review and proper link to others works; No theoretical foundation presented.

Reviewer 3 Report

Electric drives are a major part of the industry. Several control strategies based on the global trends for reducing energy consumption and energy efficiency have been developed. The introduction of electric drives reduces the energy cost. In recent years, the interest and the application of BLDC motors have increased significantly, due to their many advantages.

Following the requirements of EU standards, experts focus on the issues related to the improvement of the power factor and harmonics emission in the current of already installed electrical equipment to the power supply system.

In the present manuscript, the authors focus on standard PN-EN IEC 61000-3-2: 2019 04E. It concenrs limiting the harmonic currents injected into the public low-voltage mains electricity supply system. The manuscript also presents the results of measurements related to this standard. This shows the relevance of the manuscript.

The measuring system is, in detail, describes. The results of measurements of continuous conducted interference in the 150 kHz – 30MHz band are presented. The results of measurements of harmonics of the current drawn by the drive system, for frequencies up to 2kHz (up to the 40th harmonic component), are presented as well. The authors show multiple results. 

I would like to recommend that the authors comment in more detail what considerations should be made in order to choose the BLDC motor control techniques. What is the influence of the different BLDS motors control methods upon harmonics emission in the current? Тhe discussion of these issues is related to the fact that a compromise must be made between cost and performance when additional electronic front end circuits are added.

Round 2

Reviewer 1 Report

Dear Authors, Thank you for your response. All my questions have been answered by you. I will suggest the Editor accept the revised manuscript. Best wishes.

Reviewer 2 Report

The paper has been improved based on previous comments.